# Reversal of High-Fat Diet-Induced Non-Alcoholic Fatty Liver Disease by Metformin Combined with PGG, an Inducer of Glycine N-Methyltransferase

**DOI:** 10.3390/ijms231710072

**Published:** 2022-09-03

**Authors:** Ming-Hui Yang, Wei-You Li, Ching-Fen Wu, Yi-Ching Lee, Allan Yi-Nan Chen, Yu-Chang Tyan, Yi-Ming Arthur Chen

**Affiliations:** 1Department of Medical Education and Research, Kaohsiung Veterans General Hospital, Kaohsiung 813, Taiwan; 2Center of General Education, Shu-Zen Junior College of Medicine and Management, Kaohsiung 821, Taiwan; 3Laboratory of Important Infectious Diseases and Cancer, Graduate Institute of Biomedical and Pharmacological Science, School of Medicine, Fu Jen Catholic University, New Taipei City 242, Taiwan; 4Department of Veterinary Medicine, National Chiayi University, Chiayi City 600, Taiwan; 5School of Medicine, University of California, Davis, Sacramento, CA 95817, USA; 6Department of Medical Imaging and Radiological Sciences, Kaohsiung Medical University, Kaohsiung 807, Taiwan; 7Department of Nuclear Medicine, Kaohsiung Medical University Hospital, Kaohsiung 807, Taiwan; 8Graduate Institute of Animal Vaccine Technology, College of Veterinary Medicine, National Pingtung University of Science and Technology, Pingtung 912, Taiwan; 9School of Medicine, Kaohsiung Medical University, Kaohsiung 807, Taiwan; 10Graduate Institute of Medicine, College of Medicine, Kaohsiung Medical University, Kaohsiung 807, Taiwan; 11Department of Medical Research, Kaohsiung Medical University Hospital, Kaohsiung 807, Taiwan; 12Center for Cancer Research, Kaohsiung Medical University, Kaohsiung 807, Taiwan; 13Research Center for Precision Environmental Medicine, Kaohsiung Medical University, Kaohsiung 807, Taiwan; 14Institute of Medical Science and Technology, National Sun Yat-sen University, Kaohsiung 804, Taiwan; 15National Institute of Infectious Diseases and Vaccinology, National Health Research Institutes, Miaoli County 350, Taiwan

**Keywords:** glycine N-methyltransferase, metformin, 1,2,3,4,6-pentagalloyl glucose, nonalcoholic fatty liver disease, mitochondria

## Abstract

Nonalcoholic fatty liver disease (NAFLD) is a major cause of liver-related morbidities and mortality, and no effective drug treatment currently exists. We aimed to develop a novel treatment strategy to induce the expression of glycine N-methyltransferase (GNMT), which is an important enzyme regulating S-adenosylmethionine metabolism whose expression is downregulated in patients with NAFLD. Because 1,2,3,4,6-pentagalloyl glucose (PGG) is a GNMT inducer, and metformin was shown to upregulate liver mitochondrial GNMT protein expression, the effect of PGG and metformin was evaluated. Biochemical analysis, histopathological examination, immunohistochemical staining, reverse transcription-quantitative PCR (RT-qPCR), Western blotting (WB), proteomic analysis and Seahorse XF Cell Mito Stress Test were performed. The high-fat diet (HFD)-induced NAFLD mice were treated with PGG and metformin. Combination of PGG and metformin nearly completely reversed weight gain, elevation of serum aminotransferases, and hepatic steatosis and steatohepatitis. In addition, the downregulated GNMT expression in liver tissues of HFD-induced NAFLD mice was restored. The GNMT expression was further confirmed by RT-qPCR and WB analysis using both in vitro and in vivo systems. In addition, PGG treatment was shown to increase oxygen consumption rate (OCR) maximum capacity in a dose-dependent manner, and was capable of rescuing the suppression of mitochondrial OCR induced by metformin. Proteomic analysis identified increased expression of glutathione S-transferase mu 4 (GSTM4), heat shock protein 72 (HSP72), pyruvate carboxylase (PYC) and 40S ribosomal protein S28 (RS28) in the metformin plus PGG treatment group. Our findings show that GNMT expression plays an important role in the pathogenesis of NAFLD, and combination of an inducer of GNMT and metformin can be of therapeutic potential for patients with NAFLD.

## 1. Introduction

Nonalcoholic fatty liver disease (NAFLD) encompasses a wide spectrum of liver disorders ranging from simple hepatic steatosis to non-alcoholic steatohepatitis (NASH). Without intervention, NASH can progress to liver cirrhosis and hepatocellular carcinoma (HCC). As the most common liver disorder in the world, NAFLD is strongly linked to caloric overconsumption, physical inactivity, insulin resistance and genetic factors [1]. Given the rapid growth in patients with metabolic syndrome, the prevalence of NAFLD is expected to rise. The complex pathogenesis of NAFLD has not been fully elucidated. Accumulating evidence has indicated that mitochondrial dysfunction may contribute to the pathogenesis of NAFLD since it can alter hepatic lipid homeostasis and promotes ROS (reactive oxygen species) production and lipid peroxidation, as well as cytokine release and cell death [2]. However, the primary metabolic abnormalities leading to lipid accumulation within the hepatocytes remain poorly understood. Several potential targets and emerging pharmacotherapeutics for NAFLD are under investigation. These include metabolic-targeted therapies, oxidative-stress-targeted therapies, inflammation-targeted therapies, apoptosis-targeted therapies, fibrosis-targeted therapies, and multidrug combination therapy [3]. Although some of them are already in phase III clinical trials, these are ongoing projects and conclusions have not been made. Thus, there is no clinical approved drug therapy available so far.

Glycine N-methyltransferase (GNMT) is an important cellular enzyme regulating S-adenosylmethionine (SAM) metabolism, as well as a tumor suppressor gene for HCC [4,5,6]. GNMT has been shown to bind to polyaromatic hydrocarbons and aflatoxins, and is involved in the cytochrome P-450 liver detoxification pathways [7,8]. GNMT can also inhibit the formation of benzo[a]pyrene-DNA adducts and AFB1-DNA adducts, thus preventing their consequential cytotoxicity and carcinogenetic effect [5,8,9,10]. Interestingly, GNMT-knockout (Gnmt^−/−^) mice manifested impaired cholesterol metabolism and steatohepatitis [11]. Several putative metabolites identified in serum from Gnmt^−/−^ mice closely resemble those of NAFLD patients [12]. In addition, downregulation of GNMT has been demonstrated in the liver tissue of NAFLD patients, occurring even during early stages of chronic liver disease [13,14]. Recently, GNMT downregulation was demonstrated in liver tissues of NAFLD mice induced by a methionine and choline deficient (MCD) diet [15]. Using the MCD-induced NAFLD model, GNMT was shown to interact with and regulate complex II activity of the electron transport chain in the mitochondria of liver tissue [15]. Furthermore, recovery of GNMT levels in the mitochondria of hepatocytes showed protective effect against NAFLD [15]. Taken together, GNMT may play an important role in regulating mitochondrial function and the development of NAFLD.

Metformin (1,1-dimethylbiguanide) is the first-line medication for type 2 diabetes, with its effect primarily through improving insulin-mediated suppression of hepatic gluconeogenesis. Activation of adenosine monophosphate-activated protein kinase (AMPK), inhibition of complex I of the mitochondrial respiratory chain and alteration of cellular redox balance have been postulated to be mechanisms for gluconeogenesis inhibition by metformin [16,17]. Metformin has been considered as a promising treatment for NAFLD as studies support evidence that metformin can be a powerful therapeutic agent to reduce liver fat accumulation [18]. Furthermore, metformin was shown to upregulate liver mitochondrial GNMT protein expression in an apolipoprotein E knockout (apoE^−/−^) NAFLD mouse model [19]. Many clinical trials have been conducted to evaluate the therapeutic effect of metformin for NAFLD. However, their clinical beneficial effects are limited and conflicting reports have been published. In a meta-analysis of 417 NAFLD patients in nine clinical randomized studies, metformin was shown to improve alanine aminotransferase (ALT), aspartate aminotransferase (AST) and body mass index (BMI) to some extent, but not in histological responses [20]. In one randomized study, beneficial effects of metformin were demonstrated in body weight, serum levels of lipids and glucose, but not in liver steatosis [21]. In another randomized study in NAFLD patients receiving nutritional counseling, metformin treatment was associated with higher rates of serum aminotransferase normalization and BMI improvement than vitamin E or prescriptive diet [22]. In a double-blind randomized study with 173 biopsy-confirmed NAFLD patients, neither vitamin E nor metformin was superior to placebo in attaining sustained reduction in ALT level [23].

Previously, our group identified 1,2,3,4,6-penta-O-galloyl-β-d-glucopyranoside (PGG) as a potent inducer of GNMT from a natural plant extract library [24]. PGG is a hydrolysable tannin and a precursor of gallotannins. PGG can be purified from plants and has biological power including therapeutic potential and functional anti-microbial, anti-inflammatory, anti-carcinogenic, anti-diabetic and antioxidant properties [25]. PGG can potentially be of therapeutic usage for diseases with downregulated GNMT such as NAFLD. In this study, the effect of PGG and metformin was evaluated in a NAFLD C57BL/6J mouse model established by feeding with high-fat diet (HFD). The role of GNMT in the pathogenesis of NAFLD was also explored.

## 2. Results

### 2.1. HFD-Fed C57BL/6J Mice Showed Characteristics of NAFLD, Including Weight Gain, Elevations of Serum Transaminases and Histopathological Features of NASH

A HFD-induced NAFLD mouse model was established with C57BL/6J mice and the study design is depicted in Figure 1A. Mice were fed with either regular diet (control group) or HFD (four experimental groups) for 5.5 months, including gavage with the tested drug for the last 40 days, then were sacrificed. As shown in Figure 1B, mice fed with HFD showed significant weight gain, as well as biochemical characteristics of NAFLD including elevations of serum ALT, AST, glucose, triglyceride and cholesterol. More importantly, these HFD-fed mice developed key histopathological features of NASH in their liver tissues, including steatosis (fat droplet accumulation), lobular inflammation (foci of lymphocytic infiltrate) and hepatocyte ballooning (enlarged cells with rarefied cytoplasm) (Figure 2(A2), HFD).

### 2.2. Combination of PGG and Metformin Nearly Completely Reversed Biochemical and Histopathological Alterations of NAFLD in HFD-Fed Mice

The therapeutic effects of PGG, a potent GNMT inducer, and metformin for NAFLD were studied in the HFD-fed C57BL/6J mouse model. As shown in Figure 1B, PGG or metformin alone slightly reversed the elevated levels of serum aminotransferases and exhibited no apparent effect toward weight gain, serum levels of triglyceride or cholesterol. In comparison, combination of PGG and metformin completely reversed weight gain and elevated the level of serum transaminase (ALT) and glucose (Figure 1B).

In parallel to the reversal effects on body weight and serum aminotransferases, PGG and metformin treatments also considerably reversed the histopathological alterations of NAFLD in the liver tissues of HFD-fed mice (Figure 2A). While HFD-fed mice treated with PGG alone (Figure 2(A3), HFD+PGG) or metformin alone (Figure 2(A4), HFD+metformin) exhibited partial reversal, combination of PGG and metformin (Figure 2(A5), HFD+PGG+metformin) near completely reversed all pathological features of NAFLD. As graded by the extent of hepatic steatosis, the estimated 75% steatosis in the HFD-fed mice was considerably reduced to 50%, 35% and 5% (a 96% reduction from control) by treatment with PGG alone, metformin alone and combination of PGG and metformin, respectively. As shown in Appendix A, the combination group (HFD+PGG+metformin) had the lowest steatosis score when the control group was considered as 0. These findings indicate that a combination of PGG and metformin can effectively reverse the biochemical and histopathological alterations of NAFLD in HFD-induced NAFLD mice.

### 2.3. Combination of PGG and Metformin Reversed the Downregulated GNMT Expression and NAFLD-Related Pathologic Alterations

Immunohistochemical analysis with mouse monoclonal antibodies against GNMT was performed in the liver tissues of HFD-fed NAFLD mice. As shown, diffuse strong expression of GNMT was demonstrated in the liver tissue of regular diet-fed mouse (Figure 2(B1)). In drastic comparison, GNMT expression was greatly diminished in the liver tissue of HFD-fed NAFLD mouse, which manifested pathological alterations of NAFLD including abundant steatosis, lobular inflammation and hepatocyte ballooning (Figure 2(B2)). Interestingly, in parallel to the reversal of pathological alterations of NAFLD, the diffuse expression of GNMT was largely restored in HFD-fed mice upon treatment with PGG alone (Figure 2(B3)), metformin alone (Figure 2(B4)) or PGG and metformin combination (Figure 2(B5)).

### 2.4. Both PGG and Metformin Induced GNMT Expression in Cultured HCC Mahlavu Cells and in Liver Tissues from HFD-Induced NAFLD Mice

Quantitative reverse-transcription PCR was performed to investigate the expressions of GNMT and c-Myc in both in vitro cultured human hepatoma Mahlavu cells and in vivo hepatocytes from C57BL/6J mice. As shown in Figure 3A, PGG alone, and combination of PGG and metformin all induced significant expressions of gene and protein in GNMT and suppressed those in c-Myc (Figure 3A and Appendix A). For the group of metformin alone, the induction of GNMT was not significant, but the c-Myc was downregulated (*p* < 0.01). To mimic the NAFLD condition in vivo, Mahlavu cells were treated with oleic acid, a monounsaturated fatty acid known to induce steatosis in cultured mammalian cells [26]. As shown in Figure 3B, oleic acid treatment caused a decrease in mRNA expression of GNMT (*p* < 0.001) and an increase in mRNA expression of c-Myc (*p* < 0.001). In Mahlavu cells treated with oleic acid, similarly, PGG alone, metformin alone and combination of PGG and metformin induced significant increase in GNMT gene expression and significant decrease in c-Myc expression, with PGG and metformin combination being the strongest inducer (Figure 3B). A similar experiment was conducted in hepatocytes from C57BL/6J mice fed with regular diet or HFD. Again, all three treatments induced significant increases in GNMT gene expression, with the PGG and metformin combination being the strongest inducer (Figure 4).

### 2.5. Combination of PGG and Metformin Reduced Hepatic Steatosis and Reversed the Inhibition of GNMT

The results obtained from pathologic examination implied that PGG and metformin regulate lipid homeostasis in the liver. Therefore, specific genes related to lipid regulation were further elucidated by quantitative RT-PCR. SREBP-1c is a pro-lipogenic transcription factor that regulates the expression of genes involved in lipid synthesis. Another transcription factor is peroxisome proliferator-activated receptor gamma (PPARγ) [27,28]. HFD significantly stimulated expression of SREBP-1c and PPARγ. PGG or metformin downregulated expression of SREBP-1c but significant reduction was only observed in the group of combination treatment. Inhibitory effects of HFD-induced PPARγ expression were not observed in all the treatment groups. PPARα is responsible for β-oxidation of fatty acid in the liver [29]. Downregulated β-oxidation in NAFLD subjects leads to fatty acid accumulation in the liver, which ultimately induces ROS production and stimulates pro-inflammatory effects [30]. In this study, PPARα was decreased upon HFD administration, which was reversed significantly by PGG and combination treatment. Combination treatment inhibited the expression of HFD-stimulated specific pro-inflammatory cytokines including TNFα and IL-1β (Figure 4). These results suggested that combination of PGG and metformin inhibited hepatic lipid accumulation by reducing lipogenesis via SREBP-1c suppression, improving β-oxidation and inhibiting inflammation, and that these effects might be associated with the presence of GNMT.

### 2.6. PGG Could Rescue the Suppression of Mitochondrial Oxygen Consumption Rate Induced by Metformin

The effect of PGG and metformin on mitochondrial oxygen consumption rate (OCR) was evaluated by the Seahorse mitochondria functional analysis in both in vitro and in vivo models. As shown in Figure 5A, PGG induced an increase of OCR maximum capacity in a dose-dependent manner in cultured human hepatocellular carcinoma Mahlavu cells. PGG at 0.83 μM increased the OCR maximum capacity to 1.24-fold of the control. A 24-h treatment of metformin at 0.625 mM induced a comprehensive decrease of OCR in every condition (0 to 18.5 min, basal respiratory condition; 18.5 to 44.3 min, proton leak and non-mitochondrial oxygen consumption; 44.3 to 70.13 min, maximum respiration capacity; 70.13 to 95.94 min, non-mitochondrial oxygen consumption) in cultured hepatoma Mahlavu cells (Figure 5B). Interestingly, the comprehensive decrease of OCR induced by metformin was largely rescued by co-treatment with 0.83 μM of PGG (Figure 5B). For example, at 9.91 and 18.5 min when the model still represented the basal respiration condition of cells, the OCR induced by PGG and metformin combination was significantly higher than by metformin treatment alone (about 1.25- and 1.27-fold, respectively, *p* < 0.05).

Similar mitochondria functional analysis was conducted in the in vivo liver tissues of HFD-induced NAFLD mice. The assayed hepatocytes were derived from HFD-fed mice treated for 1.5 months with either metformin alone (150 mg/kg/day) or both metformin (150 mg/kg/day) and PGG (300 mg/kg/day). Again, the comprehensive decrease of OCR induced by metformin alone was partly rescued by PGG and metformin combination. For example, at 18.5 min, the OCR induced by PGG and metformin combination was significantly higher than by metformin treatment alone (about 1.45-fold, *p* < 0.05).

### 2.7. Proteomic Analysis

Proteomic analysis was performed for liver samples from HFD-induced NAFLD mice treated with either PGG alone, metformin alone or combination of PGG and metformin. The MASCOT analysis identified upregulation of 16 proteins, and downregulation of 10 proteins among all three treatment groups. As shown in Figure 5A, while upregulated expressions of 14-3-3 protein zeta (1433Z) and Endoplasmin (ENPL) were identified in all three treatment groups, upregulated expressions of Glutathione S-transferase mu 4 (GSTM4), Heat shock protein 72 (HSP72), Pyruvate carboxylase (PC), and 40S Ribosomal protein S28 (RS28) were only identified in the combination of PGG and metformin treatment group. On the other hand, the expressions of Fumarate hydratase (mitochondrial), Sarcosine dehydrogenase (mitochondrial) and Nesprin-3 were downregulated in all three treatment groups, and the expression of Annexin A5 and Apolipoprotein A1 was only downregulated in the combination of PGG and metformin treatment group. In the metformin alone treatment group, while 14-3-3 protein theta and cytochrome c oxidase subunit 7A2 were upregulated, protein disulfide-isomerase A3 was downregulated. In the PGG alone treatment gruop, upregulations of 3-hydroxyisobutyrate dehydrogenase (mitochondrial), cytochrome P450 and peroxiredoxin-5 (mitochondrial), as well as downregulations of acyl-CoA-binding protein and E3 SUMO-protein ligase CBX4 were identified (Figure 6).

## 3. Discussion

The liver plays a pivotal role in carbohydrate, protein and lipid metabolism. Recognized as the most common chronic liver disorder [31], NAFLD manifests major abnormalities of hepatic lipid metabolism. Currently, with no available medical treatments, the management of NAFLD relies mostly on lifestyle modification [32]. There is an urgent need to develop effective pharmacological treatment or alternative strategies for NAFLD treatment. In this regard, metformin has been considered as a promising treatment. Though clinical studies conducted so far show only limited beneficial effect, animal studies have demonstrated that metformin can be a potent therapeutic agent to reduce liver fat accumulation [18]. GNMT has been shown to be one of the top downregulated proteins in the liver tissue of NAFLD patients [13]. In addition, evidence from Gnmt^−/−^ mice and from diet-induced animal models have indicated that GNMT may play an important preventive role in the pathogenesis of NAFLD [5,11,12,13,14]. Taken together, we hypothesized that combining induction of GNMT expression with metformin may show therapeutic effect for NAFLD. In the current study, we tested the therapeutic effect of combination of PGG, a potent GNMT inducer [24], and metformin for NAFLD in a HFD-induced mouse model. Indeed, we showed that PGG and metformin combination near completely reversed all biochemical and histopathological features of NAFLD (Figure 1 and Figure 2). To our knowledge, our study is the first to show medical reversal of NAFLD by an inducer of GNMT and metformin combination.

Long-term HFD loading is known to induce obesity and insulin resistance, as well as hepatic steatosis and NASH in mice [33,34]. HFD-induced NAFLD mouse models reflect clinical disease progression and have been widely used to elucidate the pathogenesis of NAFLD, and for investigating examining therapeutic effects of drugs [33,34]. We established a NAFLD mouse model by feeding HFD, and these mice showed characteristics of NAFLD including significant body weight gain, as well as biochemical and pathological alterations of NAFLD. As shown in Figure 2A, the liver tissues of HFD-fed C57BL/6J mice manifested three key histopathological features of NASH in their liver tissues, including steatosis, lobular inflammation and hepatocyte ballooning.

Downregulated GNMT expression has been associated with NAFLD development [11,13,14,35]. Recently, downregulated GNMT expression was observed in NAFLD mice induced by a MCD diet [15]. Inversely correlating to development of biochemical and pathological alterations, GNMT expression was found to be significantly downregulated in the liver tissues of our HFD-induced NAFLD mice (Figure 2).

With downregulated GNMT expression, the HFD-induced NAFLD mice can be an ideal model system to elucidate the role of GNMT in the pathogenesis of NAFLD, and for investigating therapeutic effects of drugs. The effect of PGG, a potent GNMT inducer, and metformin on NAFLD and GNMT expression was investigated in the HFD-induced NAFLD mice. Quite effectively, combination of PGG and metformin not only reversed body weight gain and biochemical alterations, but also near completely reversed pathologic alterations of liver tissues in these NAFLD mice (Figure 1 and Figure 2). In parallel to the reversal of NAFLD-related alterations, PGG and metformin combination also successfully restored GNMT expression in the liver tissues of HFD-induced NAFLD mice. The reversal of downregulated GNMT expression by PGG and metformin was further confirmed by quantitative reverse-transcription PCR in Mahlavu cells challenged with oleic acid to mimic the in vivo NAFLD condition. Based on these findings, it is reasonable to conclude that the reversal of NAFLD-related alterations by PGG and metformin in the HFD-induced mouse model is closely linked to their combined effect to restore GNMT expression in liver cells.

Previously, GNMT was found to be the top increased protein in the mitochondria isolated from hepatocytes in apoE^−/−^ NAFLD mice treated with metformin [19]. Similarly, in our HFD-induced NAFLD mouse model, metformin treatment also induced GNMT levels in mitochondria (unpublished data). This finding indicates that metformin may promote translocation of GNMT from cytosol to mitochondria. In this regard, accumulating evidence has indicated that mitochondrial dysfunction may contribute to the pathogenesis of NAFLD [2]. In addition, previous in vitro studies revealed that metformin inhibits complex I of the mitochondrial respiratory chain [36]. Recently, GNMT was also shown to interact with and regulate complex II activity of the mitochondrial respiratory chain in a NAFLD mouse model [15]. In this study, the interesting interplay between metformin and GNMT on modulating mitochondrial function was further investigated by the Seahorse mitochondria functional analysis. We found that whereas metformin alone suppressed mitochondrial OCR, PGG co-treatment could moderately rescue the suppression of mitochondrial OCR induced by metformin (Figure 5). Taken together, our findings suggest restoration of GNMT expression by PGG and metformin-facilitated mitochondria translocation of GNMT may jointly contribute to correction of mitochondrial dysfunction, hence the reversal of NAFLD by combination of PGG and metformin.

Obesity subjects reveal increased peripheral lipolysis due to insulin resistance, which in turn generates circulating fatty acids. Fatty acids are translocated into hepatocytes and then are either stored as triglyceride via lipogenesis or oxidized in the mitochondria for ATP production. Excess hepatic fatty acid accumulation leads to ROS production which promotes inflammation and hepatic injury [30]. Combination therapy decreased serum ALT value and hepatic gene expression of TNFα and IL-1β, indicating liver injury and inflammation were reduced, respectively. This result might be correlated with regulatory effects of PGG and metformin in lipogenesis and β-oxidation.

Hepatic SREBP-1c is responsible for lipogenesis upon stimulation of insulin. The presence of TNFα also stimulates SREBP-1c, leading to steatosis [37]. PPARγ also exerts lipogenic function in the liver, and it is reported to reinforce lipogenic mechanism to SREBP-1c [38]. Our data showed that combination treatment reversed HFD-induced SREBP-1c; however, PPARγ was not affected upon treatment. The results, together with increased β-oxidation, might contribute to the phenomenon that hepatic steatosis and body weight were reduced, but hyperlipidemia was still observed. Notably, PPARγ regulates diverse cellular functions in NAFLD. PPARγ promotes the activation of anti-inflammatory M2 macrophages while reducing the number of pro-inflammatory M1 macrophages in NAFLD-associated hepatic inflammation [39], suggesting the beneficial role of PPARγ against progression of NAFLD into NASH.

Fatty acid oxidation in the mitochondria is activated by PPARα. PPARα also exerts anti-inflammatory activities in the liver inflammation [29]. Fernández-Tussy et al. showed that recovery of GNMT enhanced mitochondrial function and reduced steatosis by activating β-oxidation related genes including PPARα [15]. Therefore, metformin and PGG might contribute to enhanced GNMT expression, which results in β-oxidation and elimination of HFD-induced excess hepatic lipid accumulation.

From the view of epidemiology, NAFLD will soon become the most frequent liver disease requiring liver transplantation [40]. Thus, new protocols for the diagnosis and treatment of NASH are urgently in needed. In conclusion, our study showed that mitochondrial GNMT expression plays an important role in the pathogenesis of NAFLD, and combination of an inducer of GNMT and metformin can be an effective regimen to treat NAFLD.

In the proteomic analysis, several upregulated proteins were identified and related to energy metabolism, including gluconeogenesis and fatty acid beta-oxidation pathways. In the HFD-PGG group, three proteins, 3HIDH (3-hydroxyisobutyrate dehydrogenase, mitochondrial), CP2E1 (Cytochrome P450 2E1), and PRDX5 (Peroxiredoxin-5, mitochondrial), were upregulated and associated with liver fat metabolism; ACBP (Acyl-CoA-binding protein) was downregulated to stop the cholesterol synthesis in the hepatocyte outside of mitochondria. 3HIDH (in mitochondrion) is the last enzyme of hepatic ketogenesis and the first enzyme of ketolysis. Cytochrome P450 2E1 is involved in the metabolism of fatty acids which may catalyze the hydroxylation of carbon–hydrogen bonds. Peroxiredoxin-5 may catalyze the reduction of hydrogen peroxide and organic hydroperoxides to water and alcohols, respectively. After treatment with PGG, those three proteins may act to increase the 3HIDH expression and enhance NADP^+^ synthesized from NAD^+^. NADPH is usually used as a reducing agent for biosynthesis and cannot directly enter the respiratory chain for oxidation. Only under the action of special enzymes is the H on NADPH transferred to NAD^+^, which then enters the respiratory chain in the form of NADH. In addition, the downregulation of Acyl-CoA-binding protein in our study may decrease the fatty acid beta-oxidation of acetyl-CoA in the mitochondrial. These results may help reduce fat and glycogen accumulation in the liver and are in line with our experimental trends.

There were four proteins identified as upregulated in the two-drug treatment group only. Glutathione S-transferase Mu 4 (GSTM4) has a molecular weight of 25.5 kDa and is a protein mainly expressed in the cytosol [41]. One of its functions is to catalyze the process of forming maresin conjugate in tissue regeneration 1 (MCTR1), a bioactive lipid mediator that possesses anti-inflammatory activities. Interestingly, it was found that the symptoms of NAFLD in African Americans tend to be less progressive when compared with Caucasian NASH patients [42]. The African-American population showed upregulated expression of GSTM4, as well as GSTM2 and GSTM5. These suggest the beneficial effects of PGG combined with metformin treatment against HFD-induced liver damage.

RPS28, 40S ribosomal protein S28, is a structural constituent of ribosome and is expressed in cytosol and the rough endoplasmic reticulum (ER). RPS28 is mainly involved in ribosomal small subunit assembly and biogenesis. RPS28 seems to play roles in immunity. It has been shown that RPS28 controls MHC class I peptide generation through translation alteration [43]. RPS28 downregulated gene was causally associated with human bone marrow failure syndromes or hematologic malignancies [44]. However, its role in treating NAFLD is unclear and needs to be studied.

Pyruvate is increased in plasma and liver of NAFLD. It is converted to oxaloacetate through anaplerosis or to lactate. Increased lactate will elevate the expression of lipogenic enzymes dependent on the decreased activity of nuclear histone deacetylase. It is also increased in and contributes to the TCA cycle. The oxidation of fatty acids is also enhanced, not only in the mitochondria, but also in the endoplasmic reticulum and peroxisomes. The oxidation that occurs in the latter can lead to more production of ROS that cause inflammation and liver damage. The hepatic tricarboxylic acid (TCA) cycle provides intermediates and the energy necessary for multiple biosynthetic pathways. Pyruvate carboxylase (PC) catalyzes the carboxylation of pyruvate to OAA and represents a major anaplerotic pathway by which alanine and lactate replenish TCA cycle intermediates, not only for gluconeogenesis but also for other pathways including the urea cycle and lipid synthesis. The depletion of TCA cycle intermediates, caused by the loss of PC, imposes an oxidized NADP(H) redox state and limits antioxidant capacity. In this study, the PC was upregulated to prevent the production of ROS from pyruvate.

HSP72 is a stress-inducible and protective protein. It may be a helper during stressful conditions. To date, no studies have examined the direct effect of reduced HSP72 on hepatic lipid metabolism. However, previous reports indicate that HSP72 was significantly increased in the livers of obese mice [45]. HSP72 expression was markedly induced by palmitic acid. Furthermore, HSP72 overexpression promoted lipid accumulation in HepG2 cells. These results suggested that HSP72 may promote lipid accumulation in the early stages of NAFLD, but it can also effectively prevent high fat diet (HFD)-induced glucose intolerance and skeletal muscle insulin resistance [31,41]. Thus, induction of hepatic HSP72 may be beneficial for systemic metabolism. Hepatic HSP72 protein concentration may be adapted as a potential biomarker of NAFLD in various stages where an induction of HSP72 could be a therapeutic approach for NAFLD. Although future studies are needed to explore the therapeutic potential, this protein may play important roles in both the prevention of steatosis and the advancement of NAFLD.

The expression of Annexin A5 and Apolipoprotein A-I were downregulated in the two-drug treatment group. Annexin A5 (ANXA5) belongs to an annexin family of calcium-dependent membrane-binding proteins [46]. This protein is well known as an anticoagulant protein and has been suggested as an anti-inflammatory agent to minimize inflammation during apoptosis [47]. However, Annexin A5 expression is upregulated in a variety of cancers, such as hepatocarcinoma, cervical carcinoma, and squamous cell carcinoma, and plays roles in tumor development, invasion or metastasis [48]. Interestingly, it is also suggested that ANXA5 has interactions with lipids of the membranes [49]. Liao and co-workers adapted hamsters as an animal model due to the close similarity to human lipid metabolism. In their study, animals were fed with high-fat diet and showed upregulated expression of Annexin A5, indicating an inflammatory and procarcinogenic status induced by such a diet [50]. Thus, it is possible that the treatment of PGG combined with metformin showing downregulated expression of hepatic ANXA5 is a result of counteracting inflammatory conditions. Apolipoprotein A-I (APOA1) is a multifunctional lipoprotein and has therapeutic potential in several diseases, such as atherosclerosis, thrombosis, diabetes, cancer or neurological disorders [51]. Lipoprotein metabolism and cholesterol homeostasis are tightly linked to each other. APOA1, having a molecular weight of 30.8 kDa, is the main structural protein of high-density lipoprotein (HDL) and plays an important role in reverse cholesterol transport (RCT) [52]. In short, during RCT, the liver first synthesizes and secrets APOA1 to the bloodstream. Then, ApoA1 is transformed to HDL in which cholesterol from the peripheral tissues is transported to it with the help from LCAT (lecithin-cholesterol acyltransferase). Later, HDL transfers its cholesterol content to the liver through SR-B1 (Scavenger receptor class B member 1) [53]. Thus, we postulate that PGG combined with metformin downregulated the expression of hepatic APOA1 to slow down the cholesterol transporting from peripheral cells to liver. Yet the exact roles of downregulating expression of ANXA5 and APOA1 still need to be elucidated.

Proteomics has become an important tool for elucidating the complex pathogenesis of NAFLD. By using the HFD-induced NAFLD mouse model, proteomic analysis identified upregulation of 16 proteins and downregulation of 10 proteins among all three treatment groups. Further studies are needed to elucidate the roles of individual proteins in the process of NAFLD reversal.

PGG treatment has been shown to increase the expression of phosphorylation of AMPK (AMP-activated protein kinase) in various cells [54,55]. Although both PGG and metformin induce AMPK, their mechanisms are different [56,57]. As shown in Figure 7, PGG activates AMPK and then activates p27. On the other hand, metformin is more complicated. Metformin enters mitochondria and is involved in TCA cycle causing the increase of ROS and ATP. Caspase-9 is then activated to induce apoptosis while AMPK inhibits fatty acid synthesis.

## 4. Materials and Methods

### 4.1. Cell Cultures and Materials

The human hepatocarcinoma cell lines (Mahlavu and HepG2) were obtained from Food Industry Research and Development Institute (Hsinchu, Taiwan). Control chow diet and HFD were purchased from Bio-Cando Incorporation, Taiwan. PGG was obtained as described [24]. Metformin was purchased from BioVision (Waltham, MA, USA).

### 4.2. Animals

C57BL/6J male mice were purchased from the National Laboratory Animal Center in Taiwan. All animal experiments were performed in accordance with the guidelines and regulations for the Care and Use of Laboratory Animals and approved by the Institutional Animal Care and Use Committee of Kaohsiung Medical University (IACUC #106075). All mice were kept in a 12-h light–dark cycle room with water. Metformin was dissolved in water and PGG was dissolved in 0.5% carboxymethyl cellulose sodium salt (CMC, 419273, Sigma-Aldrich, Rockville, MD, USA). PGG (300 mg/kg-bw) and metformin (150 mg/kg-bw) were given to mice by oral gavage once a day. Mice were treated with PGG or equal volume of CMC at 10 A.M. and then treated with metformin or equal volume of water at 6 P.M. every day until sacrifice (N = 10/group).

### 4.3. Blood Biochemical Analysis

Blood samples were collected from tail artery after depriving mice of food for 12 hours. Serum samples were kept at −80 °C until assayed. Serum ALT, AST, glucose, triglyceride and cholesterol levels were measured using FUJI DRI-CHEM NX500i analyzer (Fujifilm, Tokyo, Japan) (N = 10/group).

### 4.4. Histopathological Examination, Steatosis Scoring and Immunohistochemical Staining (IHC)

Liver tissues were fixed in formalin and subsequently embedded in paraffin. The liver sections were stained by H&E staining in accordance with the standard protocol (Taiwan Animal Consortium). Steatosis score was determined according to the literature [58]. Briefly, the severity was graded based on the percentage of the affected area following the categories: 0 (<5%), 1 (5–33%), 2 (34–66%) and 3 (>66%). Detailed procedures of IHC have been reported previously [4]. A mouse anti-GNMT monoclonal antibody 14-1 (YMAC Bio Tech, Taipei, Taiwan) was used (N = 10/group).

### 4.5. Immunoblotting

Immunoblotting was performed according to our previous study [24]. Briefly, cell lysates were quantified and separated on SDS–PAGE gel, and immunoblottings were carried out using the following antibodies: anti-GNMT (14-1, YMAC Bio Tech), anti-MYC (D84C12, Cell Signaling, Danvers, MA, USA), anti-β-actin (AC-15, Sigma-Aldrich, Rockville, MD, USA) (N = 3/group).

### 4.6. Reverse Transcription-Quantitative Polymerase Chain Reaction (RT-qPCR)

Total RNA was isolated from mouse liver using TRIzol^®^ (Invitrogen; Carlsbad, CA, USA) according to the manufacturer’s instructions. RNA was reverse-transcribed to cDNA using Tetro cDNA Synthesis kit (BIOLINE; Taunton, MA, USA). The reactions were performed using KAPA SYBR^®^ FAST qPCR Kits (Kapa Biosystems; Wilmington, MA, USA). Primer sequences are listed in the Appendix A (N = 8/group).

### 4.7. Mitochondrial Functional Analysis

The Seahorse XF Cell Mito Stress Test was performed according to the manufacturer’s manual (Agilent Technologies, Santa Clara, CA, USA). Mahlavu cells (3 × 10^4^ per well) or mouse hepatocytes (2.5 × 10^4^ per well) harvested from mice receiving different treatments were seeded in the Seahorse XFp Cell Culture Miniplate (Agilent). The cells were treated with DMEM (plus 10% FBS) containing the following different reagents: 0.5 mM oleic acid (OA, Sigma Aldrich), 50 ng/mL PGG and/or 5 mM metformin. After 48 h of treatment, a supernatant of each well was then replaced with 200 µL of assay medium (DMEM without sodium bicarbonate, the pH was adjusted to 7.4). The mitochondria testing drugs, including 1.5 µM of oligomycin, 3.5 µM of FCCP and 0.5 µM of rapamycin, were loaded into the calibrated cartridge. All the tests were performed in triplicate (N = 2/Control group; N = 3/Metformin group and Metformin + PGG group).

### 4.8. Proteomic Analysis

The mouse livers were homogenized, and then the proteins were extracted by using the tissue protein extraction kit (RIPA, R0278, Merck, Rahway, NJ, USA). Protein samples (100 μL) were transferred into 1.5 mL Eppendorf tubes and incubated at 37 °C for three hours after mixing with 25 μL of 10 mM dithiothreitol (DTT, 15397, USB Corporation, Cleveland, OH, USA). The samples were reduced and alkylated in the dark at 25 °C for 0.5 h after adding 25 μL of 55 mM iodoacetamide (IAA, RPN6302V, Amersham Biosciences, Buckinghamshire, England) in 25 mM ammonium bicarbonate. Approximately 10 μL of 0.1 μg/μL modified trypsin digestion buffer (Trypsin Gold, Mass Spectrometry Grade, V5280, Promega, Madison, WI, USA) in 25 mM ammonium bicarbonate was added to the protein samples, which were then incubated at 37 °C for at least 12 h in a water bath. Two microliters of formic acid were added to each sample before mass spectrometric analysis for protein identification. The protein tryptic digests were fractionated using a flow rate of 400 nL/min with a nano-UPLC system (nanoACQUITY UPLC, Waters, Milford, MA, USA) coupled to an ion trap mass spectrometer (LTQ Orbitrap Discovery Hybrid FTMS, Thermo, San Jose, CA, USA) equipped with an electrospray ionization source. For reverse phase nano-UPLC-ESI-MS/MS analyses, a sample (2 μL) of the desired peptide digest was loaded into the trapping column (Symmetry C18, 5 μm, 180 μm × 20 mm) by an auto-sampler. Reverse phase separation was performed using a linear acetonitrile gradient from 99% buffer A (100% D.I. water/0.1% formic acid) to 85% buffer B (100% acetonitrile/0.1% formic acid) in 100 min using the micro-pump at a flow rate of approximately 400 nL/min. Separation was performed on a C18 microcapillary column (BEH C18, 1.7 μm, 75 μm × 100 mm) using the nano separation system. As peptides were eluted from the micro-capillary column, they were electro-sprayed into the ESI-MS/MS with the application of a distal 2.1 kV spraying voltage with heated capillary temperature of 200 °C. Each scan cycle contained one full-scan mass spectrum (*m/z* range: 400–2000) and was followed by three data-dependent tandem mass spectra. The collision energy of MS/MS analysis was set at 35% (N = 3/group).

### 4.9. Statistical Analysis

All experiments were performed independently and repeated at least three times. Data from the experiments are shown as mean ± standard deviation of the mean. Statistical comparisons were evaluated by one-way ANOVA.

## 5. Conclusions

NAFLD is the leading cause of chronic liver disease worldwide. Thus, new protocols for the diagnosis and treatment of NASH are urgently needed. In conclusion, our study showed that mitochondrial GNMT expression plays an important role in the pathogenesis of NAFLD, and combination of an inducer of GNMT and metformin can be an effective regimen to treat NAFLD.

## Figures and Tables

**Figure 1 ijms-23-10072-f001:**
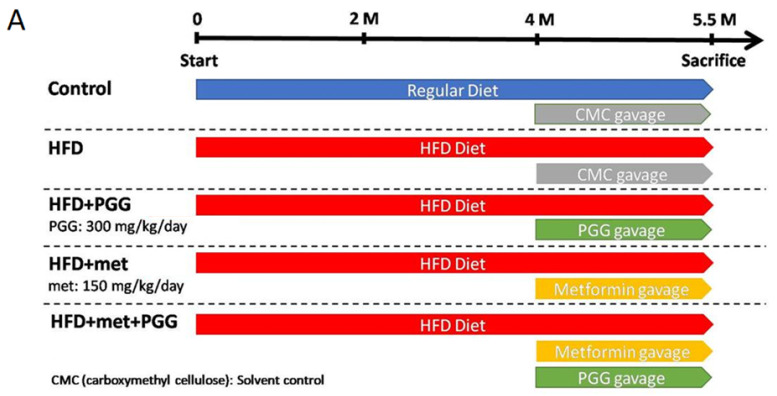
Body weight and biochemistry data of five groups of mice with indicated treatment. (**A**) The study design for the imitation of NAFLD. (**B**) Body weight and the values of serum biochemistry including ALT, AST, triglyceride, cholesterol and glucose were examined at the end point. (* *p* < 0.05; ** *p* < 0.01; *** *p* < 0.001; N = 10/group).

**Figure 2 ijms-23-10072-f002:**
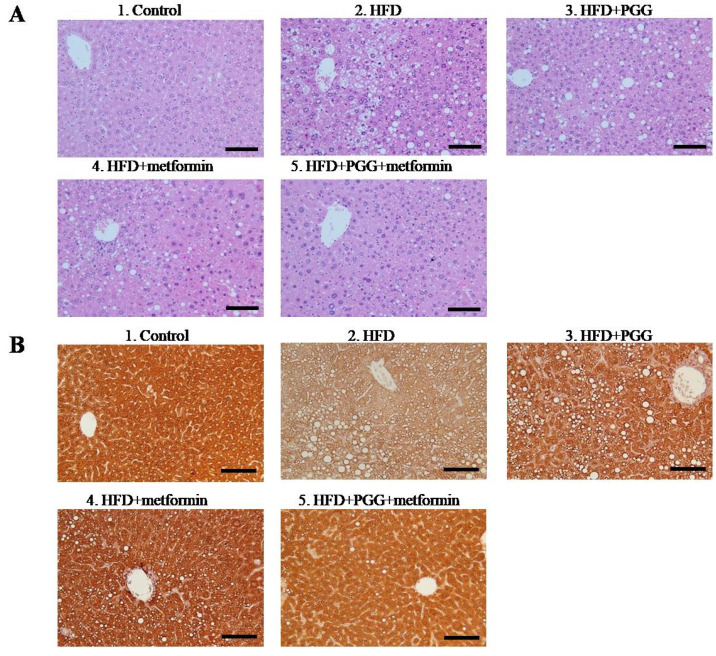
Histopathological analysis of liver sections received indicated treatment. (**A**) H&E staining. (**B**) Immunohistochemical staining of GNMT in the liver tissue. (Scale bars = 100 μm, N = 10/group).

**Figure 3 ijms-23-10072-f003:**
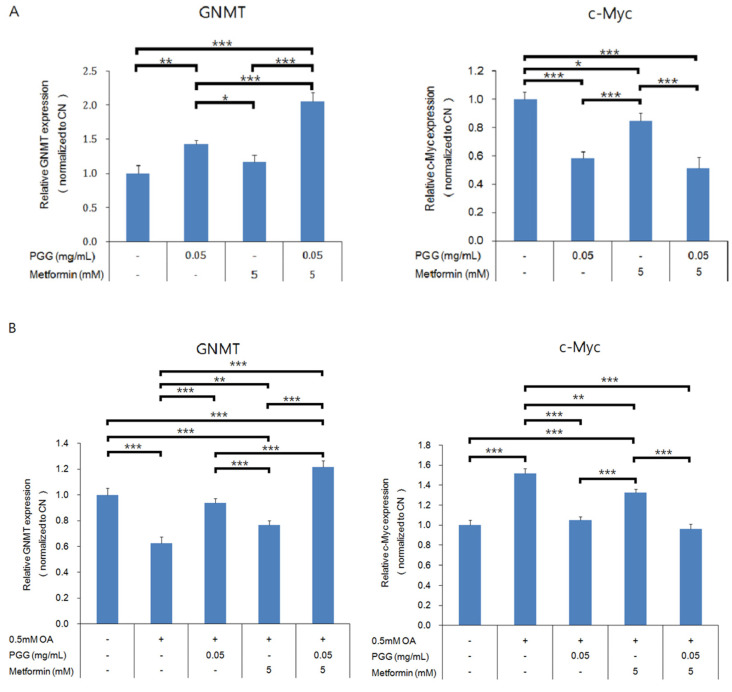
The GNMT-related gene expression in Mahlavu cells. Gene expression of GNMT and c-Myc without oleic acid (**A**) and with oleic acid (**B**). (* *p* < 0.05; ** *p* < 0.01; *** *p* < 0.001; N = 3/group).

**Figure 4 ijms-23-10072-f004:**
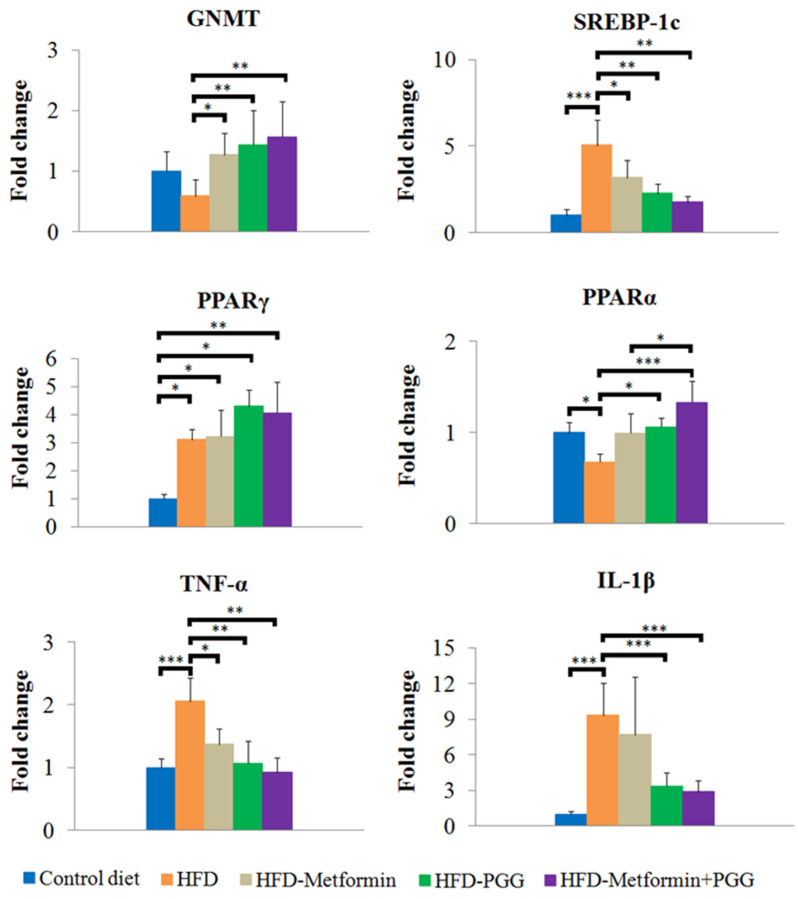
Analysis of mRNA expression for GNMT, SREBP-1c, PPARα, PPARγ, TNFα and IL-1β in the liver tissue by RT-qPCR. (* *p* < 0.05; ** *p* < 0.01; *** *p* < 0.001; N = 8/group).

**Figure 5 ijms-23-10072-f005:**
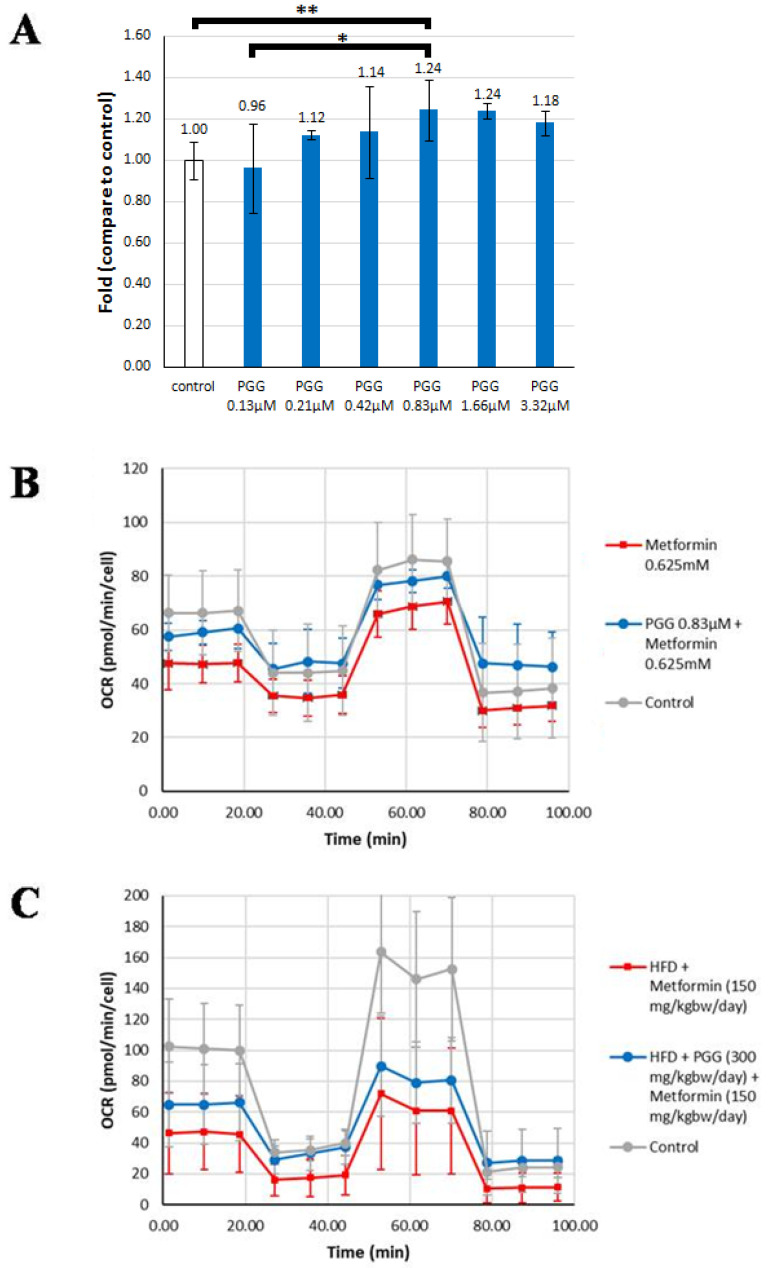
The rescue effect of PGG in metformin-induced oxygen consumption rate (OCR) decrease was evaluated by Seahorse analysis using in vitro and in vivo models. (**A**) The OCR maximum capacity in Mahlavu cells treated with different doses of PGG. (**B**) Mitochondrial respiration reflected by OCR levels was detected in Mahlavu cells treated with metformin alone and treated with metformin combined with PGG, respectively. (**C**) Mitochondrial respiration reflected by OCR levels was detected in mouse primary hepatocytes obtained from HFD-induced NAFLD mice treated with metformin alone and treated with metformin combined with PGG, respectively. (* *p* < 0.05; ** *p* < 0.01; N = 2/Control group; N = 3/Metformin group and Metformin + PGG group).

**Figure 6 ijms-23-10072-f006:**
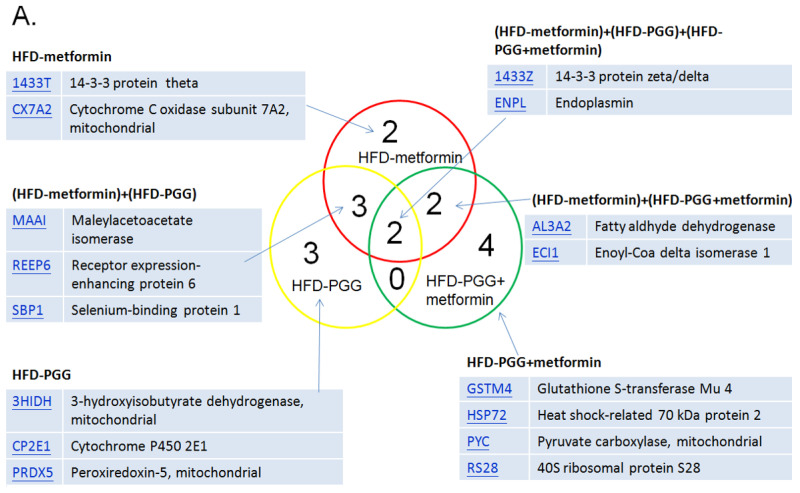
The summary of proteomic analysis results. (**A**) The MASCOT results indicate that 16 proteins were upregulated and (**B**) 10 proteins were downregulated among those three groups. (N = 3/group).

**Figure 7 ijms-23-10072-f007:**
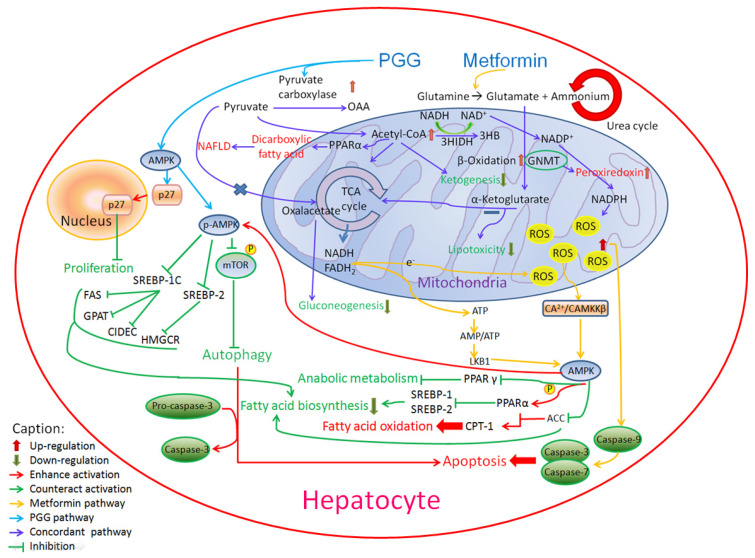
The involvement of PGG and metformin in AMPK-related pathways.

## Data Availability

Not applicable.

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
