# Peer review of "Reversal of High-Fat Diet-Induced Non-Alcoholic Fatty Liver Disease by Metformin Combined with PGG, an Inducer of Glycine N-Methyltransferase"

_ijms, 2022, doi:10.3390/ijms231710072_

Round 1

Reviewer 1 Report

This study investigated the synergistic effect of PGG and metformin in a murine NAFLD model.

Abbreviations in the abstract such as NAFLD and HFD should be defined.

Line 165, 3 x 104 > 3 × 104; Line 192, 180 um X 20 mm > 180 µm × 20 mm, and others.

Line 206, student t-test > Student’s t-test.

Amplify Figure 1A letters or words.

In Figure 2, images need bar scales, and IHC needs IgG controls.

Quality of Figure 3 also needs to be improved.

Similarly, the quality of Figure 6 also needs to be improved.

‘From the view of epidemiology, NAFLD will soon become the most frequent liver disease requiring liver transplantation (Bellentani, 2017).’ is not a conclusion.  In addition, NAFLD is the most leading cause of chronic liver disease worldwide now.

Author Response

Reviewer #1: 

This study investigated the synergistic effect of PGG and metformin in a murine NAFLD model.

  1. Abbreviations in the abstract such as NAFLD and HFD should be defined.

Reply: The authors appreciated that the reviewer pointed out this comment. The definitions of NAFLD and HFD are included in the abstract.

  1. 3 x 104 > 3 × 104; Line 192, 180 um X 20 mm > 180 µm × 20 mm, and others. Line 206, student t-test > Student’s t-test.

Reply: The authors appreciated that the reviewer pointed out this comment. These errors have been corrected.

  1. Amplify Figure 1A letters or words.

Reply: The authors appreciated that the reviewer pointed out this comment. The letters and words in Figure 1A have been amplified.

  1. In Figure 2, images need bar scales, and IHC needs IgG controls.

Reply: The authors appreciated that the reviewer pointed out this comment. Scale bars are included in Fig 2. However, we did not perform IgG controls at that time. We will keep this in mind and perform IgG control in the future studies.

  1. Quality of Figure 3 also needs to be improved. Similarly, the quality of Figure 6 also needs to be improved.

Reply: The authors appreciated that the reviewer pointed out this comment. The resolution of Figures 3 and 6 have been improved. In addition, the authors found that Figure 3C is redundant. Since the analysis of mRNA expression for GNMT is shown in Figure 4, Figure 3C (mRNA expression of GNMT) is removed.

  1. ‘From the view of epidemiology, NAFLD will soon become the most frequent liver disease requiring liver transplantation (Bellentani, 2017).’ is not a conclusion.  In addition, NAFLD is the most leading cause of chronic liver disease worldwide now.

Reply: The authors appreciated that the reviewer pointed out this comment. The first sentence of the conclusion has been modified, and the sentence of “From the view of epidemiology, NAFLD will soon become the most frequent liver disease requiring liver transplantation (Bellentani, 2017).” has been replaced by “NAFLD is the most leading cause of chronic liver disease worldwide now”.

Reviewer 2 Report

Authors correctly state that.....From the view of epidemiology, NAFLD will soon become the most frequent liver  disease requiring liver transplantation. Thus, new protocols for the diagnosis and treatment of NASH are urgently in needed and that.....Currently, besides lifestyle changes such as exercise and diet control to reduce symptoms, there is  no clinical approved drug therapies for NAFLD...quoting a reference of 2019......but they should honestly recognise that there are many drugs on the pipeline that are very good candidates to cure NAFLD/NASH as evident in various recent papers, for example...Insights into the molecular targets and emerging pharmacotherapeutic interventions for nonalcoholic fatty liver disease. Metabolism. 2022 Jan;126:154925. doi: 10.1016/j.metabol.2021.154925. Epub 2021 Nov 2. PMID: 34740573.

 Individuals with NAFLD exhibit disrupted VLDL metabolism. In fact, abnormalities in the hepatic uptake of lipoproteins and/or secretion of VLDL could lead to hepatosteatosis....Excess hepatic SAMe levels disrupt VLDL assembly and features and increase circulating VLDL clearance which will cause increased VLDL-lipid supply to tissues and might contribute to the extrahepatic complications of NAFLD as evident in...Adenosylmethionine increases circulating very-low density lipoprotein clearance in non-alcoholic fatty liver disease. J Hepatol. 2015 Mar;62(3):673-81. doi: 10.1016/j.jhep.2014.10.019. Epub 2014 Oct 18. PMID: 25457203; PMCID: PMC4336596.

What about VLDL in this model?

Authors should present their data as means plus/minus SD and not SEM because readers are interested in knowing the dispersion of values and not the precision of the mean, due to the likely paucity of observations for each group. 

By the way, it is mandatory to state the number of animals/observations in the Methods section and in every Figure.

Dealing with more that two groups, ANOVA with post-hocs should have been carried out  as appropriate statistics to better evidence the  intra and inter-group variability and draw stronger conclusions.

Author Response

Reviewer #2: 
1. Authors correctly state that.....From the view of epidemiology, NAFLD will soon become the most frequent liver disease requiring liver transplantation. Thus, new protocols for the diagnosis and treatment of NASH are urgently in needed and that.....Currently, besides lifestyle changes such as exercise and diet control to reduce symptoms, there is  no clinical approved drug therapies for NAFLD...quoting a reference of 2019......but they should honestly recognise that there are many drugs on the pipeline that are very good candidates to cure NAFLD/NASH as evident in various recent papers, for example...Insights into the molecular targets and emerging pharmacotherapeutic interventions for nonalcoholic fatty liver disease. Metabolism. 2022 Jan;126:154925. doi: 10.1016/j.metabol.2021.154925. Epub 2021 Nov 2. PMID: 34740573.
Reply: The authors appreciated that the reviewer pointed out this comment. This reference has been cited, and the manuscript has been revised as follows:
Several potential targets and emerging pharmacotherapeutics for NAFLD are under investigation. These include metabolic targeted therapies, oxidative stress targeted therapies, inflammation targeted therapies, apoptosis-targeted therapies, fibrosis targeted therapies, and multidrug combination therapy [3]. Although some of them are already in the phase III clinical trials, these are ongoing projects and conclusions are not made. Thus, there is no clinical approved drug therapy available so far.

2. Individuals with NAFLD exhibit disrupted VLDL metabolism. In fact, abnormalities in the hepatic uptake of lipoproteins and/or secretion of VLDL could lead to hepatosteatosis....Excess hepatic SAMe levels disrupt VLDL assembly and features and increase circulating VLDL clearance which will cause increased VLDL-lipid supply to tissues and might contribute to the extrahepatic complications of NAFLD as evident in...Adenosylmethionine increases circulating very-low density lipoprotein clearance in non-alcoholic fatty liver disease. J Hepatol. 2015 Mar;62(3):673-81. doi: 10.1016/j.jhep.2014.10.019. Epub 2014 Oct 18. PMID: 25457203; PMCID: PMC4336596.
What about VLDL in this model?
Reply: The authors appreciated that the reviewer pointed out this comment, but we did not measure the VLDL concentrations. We will keep this in mind and perform VLDL in the future studies.

3. Authors should present their data as means plus/minus SD and not SEM because readers are interested in knowing the dispersion of values and not the precision of the mean, due to the likely paucity of observations for each group. 
Reply: The authors appreciated that the reviewer pointed out this comment. This is a typing error. Most of the data present are already in means plus/minus SD, except Fig 5. Thus, in the revised manuscript, Fig 5 has been updated as means plus/minus SD.

4. By the way, it is mandatory to state the number of animals/observations in the Methods section and in every Figure. 
Reply: The authors appreciated that the reviewer pointed out this comment. The numbers of animals/observations in the Methods section and in every Figure have been added. 

5. Dealing with more than two groups, ANOVA with post-hocs should have been carried out as appropriate statistics to better evidence the intra and inter-group variability and draw stronger conclusions.
Reply: The authors appreciated that the reviewer pointed out this comment. Although both groups of control diet (normal diet) and high-fat diet (HFD) are shown, the statistics results focused on using the HFD group as the control. Thus, student’s t-test was applied.

Reviewer 3 Report

IJMS

The authors showed that the combination of PGG and metformin is effective in the treatment of NAFLD because it revives the expression of GNMT, whose expression is reduced in NAFLD.

The Abstract states that Western Blotting was performed, but there were no Western Blotting results in the Figures. Please show the result of Western Blotting in the Figure and Methods.

Line 34 says "The effect of PGG and metformin was evaluated." I think it would be easier to understand if you wrote the reason why you did it.

For statistical analysis of more than 3 groups, we do not use t-test. Please use other suitable methods.

In line 234, there is "(ALT and AST)", but there is no significant difference in AST (p=0.08), since it can be said that there is a significant difference only when p<0.05. Please delete.

This is important. Fig. 2A shows a picture of a liver tissue section, but this is only one representative picture out of n=10. Please measure liver TG levels and show them in Figure and perform statistical analysis, which is most important since this is a paper on NAFLD.

Figure 3 does not show results for protein expression; please illustrate the protein results or remove the word protein from the title.

PGG increases GNMT, and metformin increases GNMT even more. The reasons for this are also discussed. By the way, metformin is known to activate AMPK. Please investigate the effect of AMPK activation in combination with PGG.

The figures are small and hard to see, please make them larger.

Please read the IJMS submission rules and follow the style accordingly.

Author Response

Reviewer #3: 

The authors showed that the combination of PGG and metformin is effective in the treatment of NAFLD because it revives the expression of GNMT, whose expression is reduced in NAFLD.

  1. The Abstract states that Western Blotting was performed, but there were no Western Blotting results in the Figures. Please show the result of Western Blotting in the Figure and Methods.

 Reply: The authors appreciated that the reviewer pointed out this comment. The results of WB have been added as supplementary Figure S1, and the experimental procedure has been added in Methods.

  1. Line 34 says "The effect of PGG and metformin was evaluated." I think it would be easier to understand if you wrote the reason why you did it.

 Reply: The authors appreciated that the reviewer pointed out this comment. It has been revised as follows:
Because 1,2,3,4,6-Pentagalloyl glucose (PGG) is a GNMT inducer, and metformin was shown to up-regulate liver mitochondrial GNMT protein expression, the effect of PGG and metformin was evaluated.

  1. For statistical analysis of more than 3 groups, we do not use t-test. Please use other suitable methods.

 Reply: The authors appreciated that the reviewer pointed out this comment. Although both groups of control diet (normal diet) and high-fat diet (HFD) are shown, the statistics results focused on using the HFD group as the control. Thus, student’s t-test was applied.

  1. In line 234, there is "(ALT and AST)", but there is no significant difference in AST (p=0.08), since it can be said that there is a significant difference only when p<0.05. Please delete.

 Reply: The authors appreciated that the reviewer pointed out this comment. p=0.08 has been removed.

  1. This is important. Fig. 2A shows a picture of a liver tissue section, but this is only one representative picture out of n=10. Please measure liver TG levels and show them in Figure and perform statistical analysis, which is most important since this is a paper on NAFLD.

 Reply: The authors appreciated that the reviewer pointed out this comment. A table of steatosis scores among different groups of mice has been added as the Supplementary Table S2.

  1. Figure 3 does not show results for protein expression; please illustrate the protein results or remove the word protein from the title.

 Reply: The authors appreciated that the reviewer pointed out this comment. The word of protein has been removed.

  1. PGG increases GNMT, and metformin increases GNMT even more. The reasons for this are also discussed. By the way, metformin is known to activate AMPK. Please investigate the effect of AMPK activation in combination with PGG.

 Reply: The authors appreciated that the reviewer pointed out this comment. We have added Figure 7 to elucidate the pathways. A paragraph has also been added as the last paragraph to discuss the pathways.

  1. The figures are small and hard to see, please make them larger.

 Reply: The authors appreciated that the reviewer pointed out this comment. Figures have been enlarged.

  1. Please read the IJMS submission rules and follow the style accordingly.

Reply: The authors appreciated that the reviewer pointed out this comment. The citation and reference styles have been modified according to the IJMS submission rules.

Round 2

Reviewer 3 Report

The Student's t-test is a statistic used to test between two groups. One-way ANOVA should be used to test more than three groups. If you have two controls, one on a normal diet and one on a high-fat diet, you can compare them to both the normal diet and the high-fat diet, respectively. Retry the analysis.

In Figure 7, add informations about the genes (Srebp-1c, Ppars, etc.) targeted in your manuscript. And also indicate the pathway of fatty acid accumulation. Instead of "PGG, Metformin", please use "PGG" and "Metformin". Then, there are some that act alone and some that act in concert. Please indicate them. I think this will make this figure more effectively summarize your findings. Finally, please add a footnote to Figure 7.

Author Response

The Student's t-test is a statistic used to test between two groups. One-way ANOVA should be used to test more than three groups. If you have two controls, one on a normal diet and one on a high-fat diet, you can compare them to both the normal diet and the high-fat diet, respectively. Retry the analysis.

Reply: Thanks for reviewer's suggestion. All statistical analysis were evaluated by the one-way ANOVA.

In Figure 7, add informations about the genes (Srebp-1c, Ppars, etc.) targeted in your manuscript. And also indicate the pathway of fatty acid accumulation. Instead of "PGG, Metformin", please use "PGG" and "Metformin". Then, there are some that act alone and some that act in concert. Please indicate them. I think this will make this figure more effectively summarize your findings. Finally, please add a footnote to Figure 7.

Reply: The authors appreciated that the reviewer pointed out this comment. The genes (Srebp-1c, Ppars, etc.) were added in Figure 7. The pathway of fatty acid synthesis was indicated. "PGG" and "Metformin" were used Instead of "PGG, Metformin". All pathways were marked with different colors and the footnotes were added.